

# Carbon dynamics after five decades of different crop residue management in temperate climate

Ilaria Piccoli[1], Felice Sartori[1], Riccardo Polese[1], Antonio Berti[1]

[1]Department Agronomy, Food, Natural Resources, Animals and Environment, University of Padova, Viale dell'Università 16, Legnaro, 35020, Italy

*Correspondence to*: Ilaria Piccoli (ilaria.piccoli@unipd.it)

**Abstract.** Increasing soil organic carbon (SOC) in agricultural soils is nowadays receiving growing attention also due to the COP21 initiative of "4 per 1000". In this study, the effect of five decades of different residue management (residue removal, residue incorporation, and residue incorporation + poultry manure) was investigated on SOC stock and related to the 4 per 1000 and C saturation concepts. Preliminary results showed that higher 0-60 cm SOC stock was found after 54 years of the experiment when residues were incorporated into the soil compared to residue removal (75.0 *vs* 69.0 t ha$^{-1}$) while poultry manure had a negligible effect. Comparing the 0-30 cm SOC stock with pre-existent data series, a general decreasing trend was observed from the start of the experiment in 1966 up to 1986, being greater in residual removal (-8.6 t ha$^{-1}$) than residual incorporation (-4.8 t ha$^{-1}$, irrespective of poultry manure addition). In 2020, the difference between the above-mentioned systems was 4.1 t ha$^{-1}$ corresponding to a 2.2 ‰ which is lower than what was suggested by the 4 per 1000 initiative. This SOC stock attributed to residue retention arose in response to 141 t C ha$^{-1}$ residue resulting in a 0.1% yearly conversion rate that is sensibly lower than what is generally reported in the literature. Therefore, an alternative use (e.g., bioenergy production) of at least part of crop residues is conceivable in temperate climate for a more efficient C cycle. The studied soil was demonstrated also to be far from C saturation, being in the 30-47% degree of saturation range. Therefore, specific studies on how both organic and inorganic (i.e., carbonates) C fractions related to different soil aggregates and aggregate mineralization are namely requested.

## 1 Introduction

Nowadays the importance of increasing SOC in the soil gains growing attention due to its double function of restoring soil fertility (Tiessen et al., 1994) and mitigating climate change (Chabbi et al., 2017). As also demonstrated by the attention drawn by the COP21 initiative of *"4 per 1000"* (URL: https://4p1000.org/?lang=en) (Minasny et al., 2017) increasing and/or maintaining the SOC stock in word soils is now more than ever a central issue (Poulton et al., 2018; Rumpel et al., 2018; Soussana et al., 2019). Consequently, understanding the factors affecting and/or limiting the C sequestration into soil might play a key role in sustaining crop production, on the one side, and, protecting environmental health, on the other. Nevertheless, it is now accepted by the whole scientific community that soil has a finite capacity to store SOC and that its reaction to C input is not linear, nor infinite but tends toward a saturation level. This concept is usually defined as "carbon saturation" and was first proposed by Hassink (1997) who speculated that soil particles <20 µm (*"fines20"*) protect SOC from microbial degradation but, being fines20 finite in number, also their protection and, in turn, the amount of fine20-protected SOC is considered finite. Similarly, Dexter et al. (2008) suggested the use of clay (particles <2 µm) and a clay-to-SOC ratio = 10 to



predict phenomena of saturation. Six et al. (2002) reviewed SOC protection mechanisms and identified several protection categories: *"chemically stabilized"* for silt- and clay-associated SOC, *"physically protected"* for SOC enclosed inside microaggregate and, *"biochemically protected"* for chemically-stable SOC compounds. The same authors concluded that even if saturation level might be forecast by texture for chemically and physically protected SOC, a gap ok knowledge still exist for unprotected and biochemically protected SOC.

The carbon saturation threshold is considered soil- and land use-dependent, being greater in natural environments than in cultivated soil or increasing with the fraction of swelling clay minerals (allophanic < 1:1 < 2:1 clay minerals) (Six et al., 2002; Stewart et al., 2007). It is generally accepted that the further a soil is from the saturation, the greater its capacity and efficiency to sequester added C, whereas a soil approaching saturation will accumulate a smaller amount of SOC at a slower rate and with lower efficiency (Stewart et al., 2007).

Irrespective of saturation degree, Minasny et al. (2017) reviewed several soil management practices that can maintain or increase soil SOC levels. Among these practices, a group involved a land-use change from arable systems to pastures, perennial grasses or forests while another proposed strategy suggested the change in soil tillage management toward reduced or no-tillage practice. The former strategy implies the change from agriculture to other systems. The latter reported uncertain results on SOC sequestration potential and limited adaptability to some agroecosystems (Kay and VandenBygaart, 2002), as northern Italy (Camarotto et al., 2020; Piccoli et al., 2016), affecting the SOC stratification rather than a substantial SOC accumulation
(Baker et al., 2007; Luo et al., 2010; Ogle et al., 2012; Powlson et al., 2016). Among the strategies proposed by Minasny et al. (Minasny et al., 2017) the use of organic amendments, together with inorganic fertilizers, and previous crop residue incorporation is most likely the easier way to cope with a profitable cropping system and improve/maintain the SOC stock, at the same time. In the Veneto region agroecosystem, farmyard manure is nowadays poorly available due to the intensification of animal husbandry that produces as a main animal waste liquid slurry which presents difficulties in its transportation on long
distances, while compost is not available for extensive application. In this context, a possible alternative for organic amendment is poultry manure, as it is rich in organic C and nutrients (e.g., N), and with low water content, thus it can be more easily transported. The poultry manure incorporation, combined with crop residues, and mineral fertilizers application, appears then a suitable strategy to increase SOC (Amanullah et al., 2007; Poblete-Grant et al., 2020) and gain the 4 per 1000 objective where other types of amendments are not readily available.

Anyway, contrasting results were observed with the application of those practices in long-term experiments (LTEs) where little or no increase in SOC content was observed despite the two-to-three-fold increase in C inputs (Campbell et al., 1991; Oberholzer et al., 2014; Paustian et al., 1997; Solberg et al., 1998). Other studies observed a SOC increase due to residue incorporation with balanced mineral fertilization (Mohanty et al., 2020) or crop rotation and tillage reduction (López-Bellido et al., 2020).

Within this context, the aim of this study was to investigate the potentialities to maintain/improve the SOC stock of the two C input that are mostly available in the Veneto region i.e., crop residue and dry poultry manure using a long-term experiment started in 1966. First, the SOC and total nitrogen (TN) stocks were investigated among the 0-60 cm soil profile after 54 years of practice, second, the 2020 SOC stock evolution in the 0-30 cm soil layer was compared with pre-existent soil data series





(1966, 1982, 1986, 1993, 2006) and, finally, the degree of soil C saturation and the potentialities of reaching the 4 per 1000

goal were evaluated.

Our starting hypothesis is that both crop residue incorporation and poultry manure addition might be effective agronomic
practices to sustain the SOC stock and fulfil the 4 per 1000 goal in the temperate climate of the Veneto region agroecosystem.

## 2 Materials and methods

### 2.1 The long-term experiment

The LTE used for this study is located at the experimental farm of Padova University (Veneto Region, NE Italy 45° 21 N; 11°
58 E; 6 m a.s.l.). The climate is temperate (Cfa according to Peel et al., (2007)), sub-humid with 850 mm of annual rainfall
and temperatures rising from January (minimum average: -1.5°C) to July (maximum average: 27.2°C). Reference
evapotranspiration is 945 mm with its peaks in July (5 mm d-1). ET0 exceeds rainfall from April to September. The site has a
shallow water table, ranging from about 0.5–1.5 m in late winter/early spring to approximately 1.0–2.0 m in summer. Before
the experiment began, the land was used as a cropping area for the first decades of 1900. The trial, begun in 1966, has been
conducted on 60 35 m2 plots in a Fluvi-Calcaric Cambisol (WRB, 2014) with a silt loam texture. At the start of the experiment,
the carbonate content was measured as 33.1%, with a soil pH of 7.8, bulk density of 1.44 g cm$^{-3}$, the organic carbon content
of 1.04%, and an 8.3 C:N ratio in the 0–30 cm soil layer. The experimental treatments were derived from the factorial
combination of three crop residue management (previous crop residue incorporation "RI", previous crop residue incorporation
added with 1 t ha-1 of dried poultry manure "RI + PM", and residues removal "RR") with five levels of nitrogen fertilisation
(0, 60, 120, 180, and 240 kg ha$^{-1}$ y$^{-1}$) and four blocks. Until 1981, mineral N was applied as ammonium-nitrate, afterwards
substituted by urea. Mineral N was supplied in two top-dressing applications. In spring and summer crops, N distribution was
followed by 7-cm inter-row cultivation. All treatments received the same amounts of P (65.5 kg ha$^{-1}$ y$^{-1}$) and K (124.5 kg ha$^{-1}$
y$^{-1}$) at sowing by mineral fertilisers. The PM was applied by burying (ca. 15 cm) it during shallow disk harrowing immediately
after harvest and providing about 40 kg N ha$^{-1}$ and 410 kg OC ha$^{-1}$. Residue incorporation occurred during soil tillage in a
40/45-cm autumn ploughing and subsequent seedbed preparation (e.g., 10-cm disk harrowing). The trial was designed as a
split-plot of four blocks with residue management as the main plot and the fertilisation levels randomised inside the main plot.

Before 1984, the trial was conducted with maize (Zea mays L.) in monoculture. Thereafter, a variable rotation scheme was
used based mainly on maize, sugar beet (Beta vulgaris L.), winter wheat (Triticum aestivum L.), potato (Solanum tuberosum
L.), soybean (Glycine max (L.) Merr.), and tomato (Solanum lycopersicum L.). For a single year sorghum (Sorghum vulgare
Pers.) and sunflower (Helianthus annuus L.) were also grown. Complete information on this experiment was reported in Piccoli
et al. (2020a).

### 2.2 Soil sampling and laboratory analyses

Soil sampling was performed in 2020 at the end of the winter wheat growing season. 60 undisturbed soil cores were collected
in the middle of each plot among the 0-60 cm soil profile through a hydraulic sampler, subsequently cut into two layers (0-30
cm and 30-60 cm) and measured for bulk density according to the core method (Grossman and Reinsch, 2002). At the same
positions, also 0-30 and 30-60 cm disturbed soil samples (120 in total) were collected, air-dried, and analysed for SOC and
total Kjeldahl nitrogen (TKN). The SOC measurements were done with high-temperature catalytic combustion (SKALAR
Primacs ATC100-E, SKALAR Analytical B.V., Breda, The Netherlands) that coupled high-temperature combustion with non-
dispersive infrared detection (NDIR) to determine all the C forms, including SOC according to DIN19539.

The SOC and TKN stocks were calculated according to the equation already reported in Piccoli et al. (2016). In brief, the
equivalent soil mass (ESM) concept (VandenBygaart and Angers, 2006) was adopted in order to normalize the effects of





different bulk densities on SOC and TN stock calculation (Post et al., 2001) by applying the minimum ESM (Lee et al., 2009). The minimum ESM applied were 3798, 4190 and 7988 t ha$^{-1}$ for 0-30, 30-60 and 0-60 cm soil layers, respectively.

**2.3 Data elaboration and statistical analyses**

Data belonging to the 2020 sampling campaign were analysed with a linear mixed-effect model based on a restricted maximum likelihood estimation method treating crop residue management, N level, and their interaction as fixed while the block as a random effect. Post-hoc pairwise comparisons of least-squares means were performed, using the Tukey method to adjust for multiple comparisons. Statistical analyses were performed with SAS software (SAS Institute Inc. Cary, NC, USA), 5.1 version.

The 0-30 cm SOC stock obtained in 2020 was then compared with the pre-existent soil data series (1966, 1982, 1986, 1993, 2006) in order to evaluate the SOC evolution. From 1966 to 1993 SOC was measured with the dichromate oxidation method (Walkley and Black, 1934) while in 2006 with flash combustion using a CNS Elemental Analyzer (for further detail, please see Poeplau et al., (2017)) after the removal of inorganic C with acid pretreatment. When the analytical technique changed, tests were carried out by our laboratory to ensure that data from the two techniques were compatible.

The same data series was used to estimate the degree of soil C saturation according to the model proposed by Hassink (1997), Dexter et al. (2008) and a few selected from Six at al. (2002). In brief, the hereafter called *"Hassink"* and *"Dexter"* models estimates the maximums storable SOC as 1/20 of the fines20 (i.e., particle < 20 µm) and 1/10 of the clay (i.e., particle < 2 µm), respectively. Among the 11 regression models proposed by Six at al. (2002) we tested the ones more suitable for the studied experimental design, i.e., the models involving a cultivated ecosystem with prevalence of 1:1 clay mineral considering the protection capacity of both particles < 20 and < 50 µm.

Residue biomass was measured for each plot during harvest. Endogenous SOC from belowground crop roots, including rhizodeposition, was estimated at 1/3 of the aerial biomass (Johnson et al., 2006). For all plant tissues, an average C concentration of 0.45 g C kg$^{-1}$ (dry matter) was assumed.

**3 Results**

**3.1 SOC concentration, TN concentration and C/N ratio**

In 2020 the SOC concentration showed significant differences according to the treatment at both upper (0-30 cm) and deeper (0-60 cm) horizons. In the 0-30 cm soil layer, the treatment involving residue incorporation (i.e., IR and IR+P) significantly (p<0.001) resulted in greater SOC concentration of about 0.10 g 100 g$^{-1}$ (0.93 vs 0.84 g 100 g$^{-1}$) (Figure 1-a). The same trend occurred also at the deeper soil layer (30-60 cm), with greater SOC concentration under RI and RI+PM (0.83 g 100 g$^{-1}$, on average) compared to RR (0.77 g 100 g$^{-1}$) (Figure 1-b). The N fertilization rate significantly (p=0.03) affected the SOC concentration only in the upper layer with the greatest (0.93 g 100 g$^{-1}$) and lowest (0.86 g 100 g-1) values at 120 and 0 kg N ha$^{-1}$, respectively. The SOC concentration was unaffected by the N fertilization rate at subsoil with an average content of 0.81 g 100 g$^{-1}$.

The TN content was significantly (p=0.01) affected only by the residue management in the 0-30 cm layer with similar TN in RI and RI+PM (0.094 g 100 g$^{-1}$, on average) that was greater than RR (0.085 g 100 g$^{-1}$) (Figure 1-c). In the 30-60 cm layer,



neither the residue management nor the N application rate affected the TN concentration, and the mean value was 0.083 g 100 g$^{-1}$ (Figure 1-d).

The C-to-N ratio was not significantly affected by residue management nor by N application rate at both studied layers, ranging from 8.7 to 13.2 (average: 9.9) at topsoil and from 8.5 to 12.2 (average: 9.8) at subsoil (Figure 1-e,f).

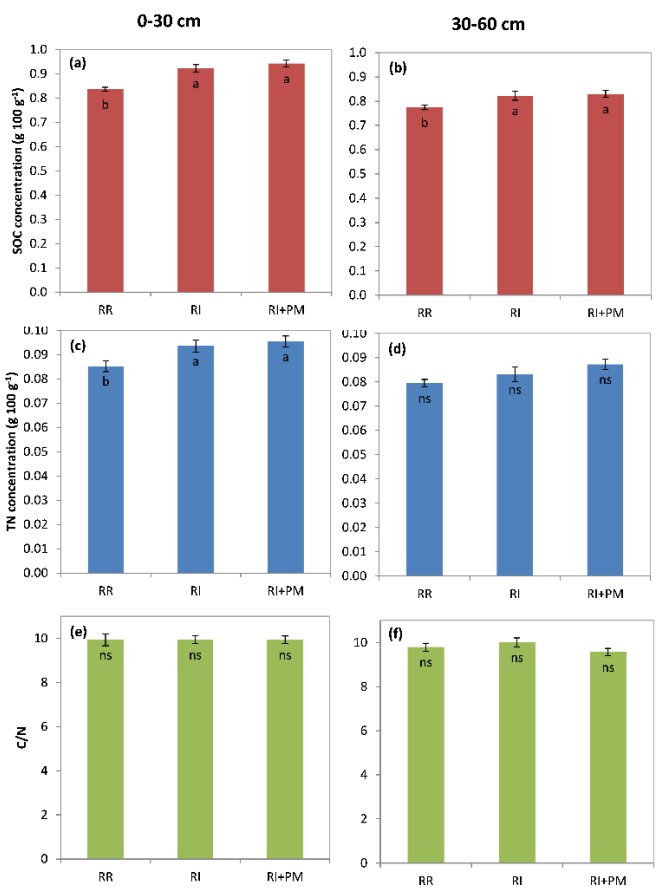


**Figure 1: Soil organic carbon (SOC) (a, b), total nitrogen (TN) concentration (c, d) and C/N ratio (e, f) in the 0-30 (a, c, e) and 30-60 cm (b, d and f) soil layer. Different letters indicate differences according to the Tukey post hoc test at p<0.05. ns= not significant. The bars indicate the standard error. RI+PM: residue incorporation + poultry manure, RI: residue incorporation, RR: residue removal.**

**3.2 The SOC and TN stocks**

The 0-30 cm SOC stock followed the same trend observed for the SOC concentration with a significant (p<0.001) greater stock in RI and RI+PM (38.7 t ha$^{-1}$, on average) compared to RR (34.7 t ha$^{-1}$) (Figure 2-a). Similarly, also subsoil receiving residues (both RI and RI+PM) had a SOC stock significantly (p=0.02) higher than subsoil without residues (i.e., RR) of about 2.0 t ha$^{-1}$ (36.2 vs 34.3 t ha$^{-1}$) (Figure 2-a). The N application rate affected (p=0.005) the SOC stock in the 0-30 cm layer with 155 the 180 kg N ha$^{-1}$ rate observing the greatest (38.7 t ha$^{-1}$) and the 0 and 60 kg N ha$^{-1}$ the lowest stock (35.9 t ha$^{-1}$, on average). On the contrary, the N application rate did not affect SOC stock at the deeper soil horizon with an average of 35.6 t ha$^{-1}$ across all the N application rates. Overall, the 0-60 cm SOC stock was significantly affected by both residue management and N





application rate being the residue incorporation treatments (75.0 t ha⁻¹, on average) greater than the RR (69.0 t ha⁻¹) and the plots with 180 kg N ha⁻¹ application rate higher than 0 and 60 kg N ha⁻¹ ones (75.3 vs 70.6 t ha⁻¹).

The TKN stock was significantly affected by residue management at topsoil following the same pattern observed for SOC stock with RI and RI+PM having a greater stock than RR (3.6 vs 3.2 t ha⁻¹) (Figure 2-b) but with no N application rate effect. The 30-60 cm layer was unaffected by both residue management and N application rate with an average stock of 3.5 t ha (Figure 2-b). The overall TKN stock across the 0-60 cm soil profile was significantly (p=0.01) affected by residue management (RI and RI+PM > RR) without any fertilizer rate effect (Figure 2-b).

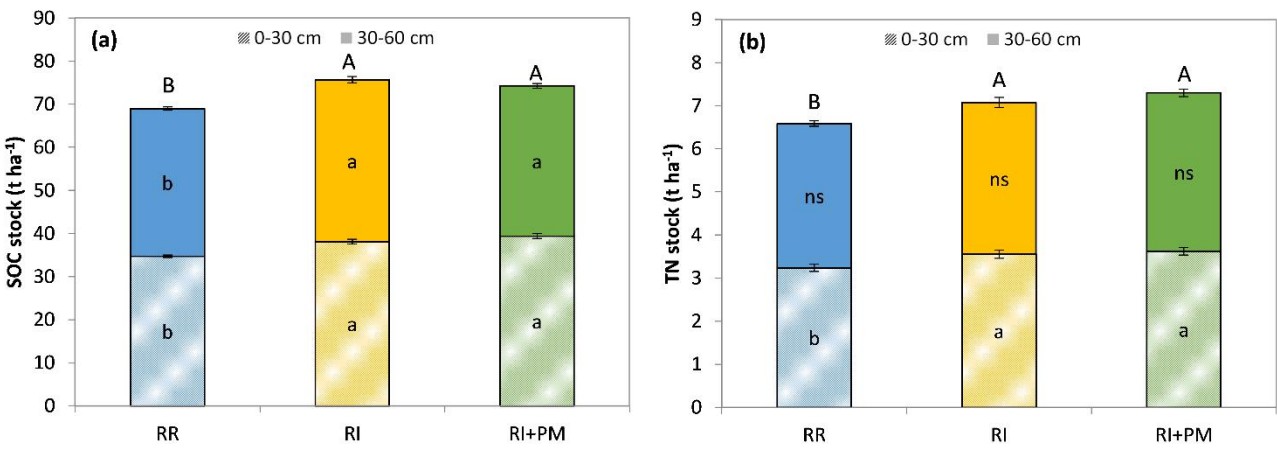

**Figure 2: Soil organic carbon (SOC) (a) and total nitrogen (TN) stock (b) among the studied soil profile. Different letters indicate differences according to the Tukey post hoc test at p<0.05 where lower case letters represent the single layer (0-30 and 30-60 cm) while upper case letters the entire studied soil profile (0-60 cm). ns= not significant. The bars indicate the standard error. RI+PM: residue incorporation + poultry manure, RI: residue incorporation, RR: residue removal.**

**3.3 Soil C saturation**

The soil C saturation exhibited a declining trend from the staring of the experiment until now in all tested models. Higher saturation level was estimated by Six (1:1 0-20 µm) model (average: 47%) followed progressively by Six (cultivated 0-20 µm) (average: 45%), Dexter (average: 41%), Six (cultivated 0-50 µm) (average: 39%), Six (1:1 0-50 µm) (average: 34%) and Hassink model (average: 30%). Despite some differences in the absolute C saturation degree, the variation between the start

of the experiment and 2020 always amounted to about 19%.

**4 Discussion**

**4.1 SOC stock and its evolution**

After 54-yr of different crop residue management, residue incorporation showed higher 0-60 cm SOC stock compared with residue removal (75.0 *vs* 69.0 t ha⁻¹), irrespective of poultry manure addition. The SOC stock was almost equally divided into

0-30 (51%) and 30-60 cm layers (49%). Nevertheless, the equal distribution of SOC among the soil profile evidenced on the one hand the great potentialities of deep soil horizon to stock SOC and, on the other, the importance of sampling deeper into





the soil in order to correctly evaluate the effectiveness of agronomic practices on C cycle (Chenu et al., 2018; Dal Ferro et al., 2020; Lal, 2018; Morari et al., 2019).

A general decreasing trend was observed from the start of the experiment up to 1986 in the 0-30 cm soil layer, being greater in residual removal (-8.6 t ha⁻¹) than residual incorporation (-4.8 t ha⁻¹, irrespective of poultry manure addition) (Figure 3). Afterwards, the gap between RR and RI/RI+PM maintained stable at about 4.1 t ha⁻¹. Similar decreasing trends occurred in other European LTEs, as revealed by Shahbaz et al. (2019) in Swedish and by Xu et al. (2021) in Belgian soils. The former authors observed how also high C input levels were not sufficient to counteract SOC losses during the last six decades due to the greater availability of freshly added C for microbial degradation compared to old C. This was particularly evidenced under

mineral fertilized rather than unfertilized plots and was explained by the authors as the results of the priming effect (Shahbaz et al., 2019). The latter (Xu et al., 2021) partially explained the SOC loss in Flanders as the result of the land-use conversion from permanent pastures to cropland with the prevalence of maize cropping (Sleutel et al., 2007).

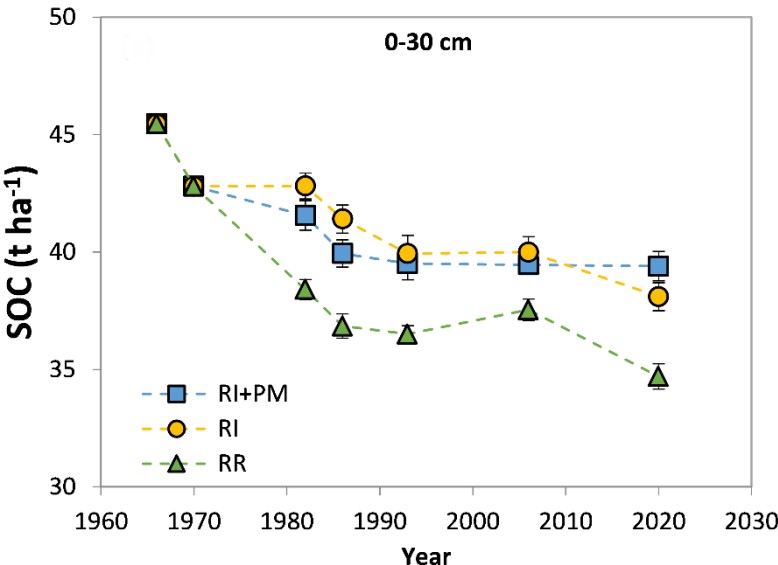

**Figure 3: The 0-30 cm SOC stock evolution from the start of the experiment (1699) up to the present. RI+PM: residue incorporation**
**+ poultry manure, RI: residue incorporation, RR: residue removal.**

The experimental farm where the LTE is located was owned by the University by 1962. Before the start of the experiment, the field was conducted following the intensification of agricultural practices, typical of the first part of the 60s. It can then be hypothesised that at the beginning of the experiment the system was not at equilibrium, as confirmed by the low C/N ratio in 1966. Therefore, the general SOC decline might not represent the effect of a land-use change but rather a movement from a

threshold level to another inside a cultivated agroecosystem, in particular a shift from a low (e.g., shallow non-inversion tillage) to more intensive agriculture (e.g., moldboard ploughing). Steward et al. (2007) previously speculated that a soil C pool can stabilize at different SOC levels due to different agronomic-derived soil disturbances (e.g., soil tillage operations). Moreover, it is also conceivable that, along with the tillage intensity, the cropping area was up to the 60s characterized by greater crop diversity, including fallow period or alfalfa, and the addition of organic fertilizers that might have sustained the SOC at higher

levels (López-Bellido et al., 2020; Morari et al., 2006). Contrarily to those findings, Liao et al. (2015) found that agricultural intensification being characterized by not only high agricultural mechanization but also by greater use of agronomic input (e.g., fertilizers and pesticides), resulted in a significant increase in crop productivity –i.e., greater agricultural products and crop residues- that, in turn, fostered greater topsoil (0-20 cm) SOC stock. Nevertheless, the Chinese experimentation covered three decades and might show saturation-related phenomena in the next decades. Indeed, in the long-term the maintenance of





adequate SOC levels in arable soils might require a huge quantity of fresh organic residues, up to 10 times what it is eventually expected to sequester (Berthelin et al., 2022), making this practice almost practically impossible to achieve.

**4.2 The SOC and TN stocks**

The gap between residue incorporation treatments and RR was about 4.1 t ha$^{-1}$ in 2020 corresponding to a 2.2‰ yearly difference between residue removal and incorporation. Anyway, in all the treatments, the SOC decreased not allowing it to
reach the target of the 4 per 1000 initiative (https://www.4p1000.org/). The "4 per 1000" initiative aspires to increase soil organic matter stocks by 4 ‰ per year as compensation for the global emissions of greenhouse gases by anthropogenic sources (Minasny et al., 2017) but its practical effect is still under debate (Minasny et al., 2018; de Vries, 2018; White et al., 2018). These about 4 t SOC ha$^{-1}$ attributed to residue retention resulted in response to 141 t ha$^{-1}$ of residue-derived C. Thus, a 2.9% conversion rate from residue to SOC stock resulted during the 54-yr experimentation. These values are sensibly lower than
what is usually reported in the literature. For example, Berti et al. (2016). Barber (1979) and, more recently, Xu et al. (2021) estimated an 11% conversion rate of the residue-derived C into new SOM. The lower performances of studied soil in terms of crop residue humification might be the result of a higher mineralization rate. Unfortunately, to the best of our knowledge, no specific studies have been conducted on this topic in the Veneto region pedoclimatic conditions.

Nevertheless, as previously postulated by Poeplau et al. (2015) an alternative use of crop residue might be energy production.
The heating value of 1 mg of residue corresponds to 0.33 mg of diesel (Lal, 2005; Larson, 1979), therefore the annual replacement of 2.6 t C ha$^{-1}$ y$^{-1}$ would have saved 0.86 t diesel ha$^{-1}$ y$^{-1}$ which corresponds to ca. 0.74 t C ha$^{-1}$ y$^{-1}$. This means that using crop residue for bioenergy production rather than for SOC sequestration might save ca. 10-fold more $CO_2$-equivalents than SOC accumulation by incorporation.

In RI+PM, the incorporation of 1 t ha$^{-1}$y$^{-1}$ of poultry manure significantly affected neither the 0-30 cm SOC stock nor the TN
stock, in comparison to the RI despite a cumulative addition of about 22.1 t OC ha$^{-1}$ and 2.2 t OM ha$^{-1}$. These results contrast with previous studies where a positive effect on SOC stock was revealed (Amanullah et al., 2007; Poblete-Grant et al., 2020). As for crop residues, these results could be linked to the high mineralization rate observed in the studied soil, where the addition of nitrogen did not contribute to SOC stock increase, while positively affecting plant nutrition (Piccoli et al., 2020a).

**4.3 Testing the soil C saturation concept**

According to the *"carbon saturation concept"* the SOC stock on a specific soil is expected to stabilize around a finite value directly linkable to the fine soil fraction, i.e., silt and clay fractions. Hassink (1997) and Dexter et al. (2008) reported two models to predict saturation phenomena by relating the SOC stock to fines20 and clay content, respectively. According to these models, a fine20-to-SOC ratio = 20 and a clay-to-SOC ratio = 10 might represent threshold values that were recently also confirmed in the long-term experiment at Rothamsted Research (Jensen et al., 2019). Six et al. (2002) reviewed the
saturation concept and developed a few models that considers also the ecosystem (e.g., cultivated vs forest) and clay type (e.g., 1:1 vs 2:1). The Veneto region studied soil presented an average clay and fine 20 content of about 24 and 66 % respectively. Comparing the actual SOC stock to saturation models, only a partial soil carbon saturation was evidenced. Indeed, the studied soil was demonstrated to be far from saturation, being in 2020 between 27 (Hassink) and 43% (Six 1:1 for 0-20 µm size) degree of saturation, according to the adopted method (Table 1). Therefore, Veneto region silty soils are theoretically able to
complex-bind larger quantities of SOC according to tested models. Nevertheless, Dal Ferro et al. (2020) recently showed that Padova University LTEs are far from Dexter-derived saturation in tilled soils including the plots where also high levels of organic fertilizer (e.g., farmyard manure) are applied. The same authors confirmed that a permanent meadow that is not tilled was the only cropping system able to approach the clay-to-carbon ratio equal to 10 and, at the same time, reach the "4 per 1000" goal. The above-mentioned meadow observed a Dexter saturation of 82% on average but showed a decreasing trend
from 1974 (92%) to 2012 (76%) (data not shown). Short-term experiments in the Veneto region previously demonstrate how the studied calcaric silty soil is inert to agronomic management change (Camarotto et al., 2018), suffering from compaction



issues and anoxic conditions (Piccoli et al., 2017, 2020b), and not increasing SOC stock (Camarotto et al., 2020; Piccoli et al., 2016).

**5 Conclusion**

The present LTE study highlighted how the SOC stock in the Veneto region agroecosystem declined as a result of agricultural intensification and seemed to stabilise to equilibrium value only after 16 years. The adoption of crop residue incorporation allowed a lower magnitude of SOC decline with respect to residue removal practice while the addition of poultry manure did not show any relevant effect. The residue rate of conversion (C from residue into SOC) was sensibly lower with respect to other European LTE and was even not sufficient to reach the 4 per 1000 goal. Therefore, an alternative use (e.g., bioenergy

production) of, at least part, crop residues is conceivable in a temperate environment for a more efficient C cycle. However, other beneficial effects of straw incorporation for soils, such as structural improvement, soil erosion reduction and nutrient recycling, must be considered.

Finally, it is speculated that studied soils are far from C saturation according to the main available models, but they show slow C dynamics. It is thus possible that the high carbonate contents (ca. 33%) mainly concentrated in the fine20 fraction might

hinder the benefits of soil C input in tilled fields. Therefore, specific studies on both organic and inorganic (i.e., carbonates) C content as they related to different soil aggregates are namely requested to properly evaluate the applicability of the carbon saturation concept to a wider set of soil conditions.

**Data availability statement.** The data that support the findings of this study are available from the corresponding author upon reasonable request.

**Author Contributions.** Formal analysis, I.P. and A.B.; investigation and data curation, F.S. and R.P.; writing—original draft preparation, I.P.; writing—review F.S, I.P., R.P. and A.B.; visualization, I.P.; supervision, A.B. All authors have read and agreed to the published version of the manuscript.

**Funding statement.** The research leading to these results has received funding from the Italian Ministry of Education, University and Research (MIUR) through several PRIN Projects (Grants Nos. 2002071492, 2005071990, 2007J5Z9LK, and
2010FRE7J4).

**Conflict of interest disclosure.** Authors declare any conflicts of interest.

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
