# Peer review of "Carbon dynamics after five decades of different crop residue management in temperate climate"

_EGUsphere, 2022_

## Author Comment (AC1)

This paper reports soil organic carbon (SOC) concentrations and stocks from a long-term residue, manure, and fertilization experiment. The authors discuss the changes in SOC between residue and manure treatments and over time in the context of soil carbon saturation and the '4 per mille' recommendation, providing the hypothesis that crop residue incorporation and poultry manure addition may be effective for fulfilling goals of '4 per mille'. While most of the data are novel and the general concept of reporting SOC levels in a long-term field experiment is sound, I have multiple concerns surrounding almost all aspects the of the sample collection, discussion of experimental design, data reporting, analytical methods, interpretations of results, and writing style.

We thank the reviewer for appreciating the amount of work and people that are behind this type of long-term experiments and for pointing out these issues that were promptly considered for manuscript improvement.

Specific comments:

 Lack of clarity about number of cores taken per experimental plot for SOC analysis. The methods section is written in a way that it is not entirely clear how many soil cores were taken per plot, but it seems possible that there were 60 'disturbed' cores taken in total, one for each experimental plot? If so, this is a strong limitation of the study. SOC is highly spatially heterogenous, so if one core per plot was sampled, even with 4 plot-level replicates of each treatment, this coring strategy would introduce likelihood that true differences in SOC levels due to treatment were not clearly observed and that reported differences reflect a large element of spatial heterogeneity across the field trial rather than showing treatment effects. If one core per plot was taken, while the sampling strategy cannot be altered at this stage, I would strongly suggest to the authors to clearly report the number of cores taken per plot for SOC analysis (e.g., "We sampled one core per plot for SOC analysis, and divided it into two depths") rather than only total number of cores, and to justify this approach, and to acknowledge in the discussion section the limitations introduced into the detection of treatment differences by this likely under-sampling of the field trial.

- We got the point of the reviewer, and we agree. We will modify the entire sampling section clarifying the sampling procedure. In brief, the soil sampling involved two different samplings. 1) 60 undisturbed soil cores (7 cm diameter, 60 cm height) were collected from the middle of each plot using and hydraulic sampler, cut into distinct layers (0-30 and 30-60 cm) and oven-dried for bulk density determination. 2) Disturbed soil samples were collected from 5 positions inside each plot in the 0-30 cm and 30-60 cm layers using a hand-push auger. Afterwards, the 5 sub-samples per plot per depth were mixed to form 120 (60 plots x 2 depths) soil samples.

- Confounding of tillage with residue retention in experimental design. The experimental design consisted of two treatments of residue incorporation (one with manure) and another with residue removal, at all levels of N fertilization, where only the treatment of residue removal was not disturbed, thus presenting

a confounding effect of tillage with residue. Tillage can affect amount and distribution of SOC. While this long-term treatment is not under the control of the authors, the rationale for the experimental design, its drawbacks, and the ambiguity it introduces into interpretation of the results should nevertheless be carefully discussed.

- The tillage practices were the same across the entire experimental area meaning that all treatments, including residue removal, received the same tillage operations. We will clarify this aspect in the text by stating that "the tillage operations were the same in all treatments, consisting in soil ploughing followed by rotary harrowing".

- Use of different analytical methods to measure SOC over time. The authors report pre-existing data from a time series of SOC sampling at the site, where earlier samples were assessed with the dichromate oxidation method and later samples were analyzed with a flash combustion method. Different analysis methods for SOC return different results for the same soils (Roper et al. 2019 https://doi.org/10.2136/sssaj2018.03.0105); while the authors performed a methods comparison with this in mind, the results of this comparison are not reported (paragraph line 115). I suggest that the authors report summary statistics and show data, possibly in supplementary materials, of their methods comparison tests and also incorporate a discussion of how the interpretation of time series data may be affected by variability and especially bias introduced by the different SOC analysis methods.

- We got the point of the review, and we agree in considering the analytical methods a thorny subject especially when dealing with long-term experiments. She/he will probably already know that all the research groups running LTE spend a lot of effort in keeping the data comparable across different years. Up to 1994, the dichromate oxidation method was used to analyse SOC concentration. In 2006 flash combustion (Elemental Analyzer) was introduced, and, in this study, we used the novel high-temperature catalytic combustion according to DIN19539. For the first analytical technique change, we obtained a slope of 0.99025 while 0.9221 for the second method change. Here below you can see the related summary statistics of those regressions. We agree that this is a central issue, and we can add the regression in the supplementary material of the final version of the paper. Moreover, in the discussion section, we can add a paragraph speculating on how different analytical techniques might have affected our results.

| Flash combustion vs dichromate oxidation | High-temperature catalytic combustion vs flash combustion |
|---|---|
| Call: | Call: |
| lm(formula = Conc ~ 0 + WB) | lm(formula = Skalar ~ 0 + CNS) |
| | |
| Residuals: | Residuals: |
|    Min    1Q  Median    3Q    Max |    Min    1Q  Median    3Q    Max |
| -0.45384 -0.17619 -0.07775 0.06149 1.28087 | -0.113312 -0.030874 -0.004654 0.040663 0.120899 |

- Non-reporting of effect of N fertilization on SOC stocks. While the authors only report SOC levels per residue and manure addition treatment, there were five levels of N fertilization also sampled. In Figures 1-3, which N fertilization levels are represented? Or, are N fertilization levels averages across residue and manure treatments? Please specify in the methods and in the figure legends.

- We did not focus much on N since only a few significances were evidenced. Anyway, we understand the comments of the reviewer and we will add more information regarding the N-level effect on SOC. In Figg 1-3 the average of all N levels is reported. We will improve the figure legends, as suggested. Moreover, we will add a new figure showing the effect of N on SOC concentration, where significant.

- Non-reporting of C input estimates. Although a calculation of C inputs is discussed in the methods section and reported in the abstract, the data on C inputs across treatments are not currently reported in any table, or figure, or in the results section. Please add these results.

- We appreciated the reviewer's comment which will be promptly considered for MS improvement by adding text, figures/table on C input.

- Absence of results section corresponding to Figure 3 (SOC change over time). The authors present Figure 3 in the discussion section, and these data are discussed in the abstract and conclusions, but currently there is not a results section corresponding to this figure. I suggest the authors either expand existing results sections to include results reporting for this figure, or create a new results section.

- We got the point of the reviewer, and we will move Fig. 3 to the results section and will expand this part by also adding the corresponding text.

- Overall lack of legibility of data visualization. Particularly in Figure 2, the shadings used for different depths can't be discerned from the legend. The caption in Figure 2 doesn't specify if post-hoc comparisons represented by letters are across treatments within the same depth; this seems likely but could be clarified (same in Figure 1). In Figure 1, larger font size for axis labels would improve readability.

- We will revise the Figure by increasing the readability, as suggested.

- Claim of testing the soil carbon saturation concept: the authors use previous literature to calculate the expected maximum of soil organic carbon in mineral-associated form (MAOC) based on soil texture at the experimental site, and reasonably point out that the SOC levels observed were below the expected theoretical maximum, and therefore far from saturation. While their conclusion is probably sound, there are several conceptual and analytical flaws with the work, and a more nuanced approach and discussion would improve the rigor of the interpretation and claims. First, the concept of soil carbon saturation applies specifically to the mineral-associated carbon (MAOC), isolated through soil disturbance and size or density cutoffs. However, the authors measure and report only total soil organic carbon, which also includes particulate organic carbon (POC) that is not theorized to be controlled by saturation limits. Since the SOC (=MAOC + POC) reported is still below the theoretical maximum of MAOC based on soil texture, the claim that the soils were below saturation (based on this theory and method accounting) is still accurate, but it needs to be acknowledged the implications of the presence of POC in the sample and how this affects the saturation estimate. Second, the authors use older references for their calculation of MAOC at saturation; why use these rather than larger and more recent datasets? E.g., Feng et al. 2013 1007/s10533-011-9679-7; Georgiou et al. 2022 10.1038/s41467-022-31540-9.

- We got the point of the reviewer, and we agree. Regarding the first raised point we will specify that the saturation concept applies to MAOC and, therefore, we will include a paragraph in the discussion speculating about the possible implication of the presence of POC in the sample and how this affects the saturation estimate. We thank the reviewer for suggesting two interesting papers. We tried to apply the suggested relation presented by Feng et al. which refers to cultivated soils and gave a slightly higher saturation level. Anyway, as pointed out by the Reviewer the soils are far from saturation even including POC and this will be one main point for subsequent discussion.

- Interpretation of 4 per mille objective based on last SOC sampling only, when all treatments studied appear to decrease in SOC over time. The efficacy of 4 per mille and other natural climate solutions depends on increasing SOC levels, so the results of the time series, so long as they are based on sound inter-method comparison, would represent a repudiation of the 4 per mille at this site.

- We understand your concern and we partially agree. Indeed, if on the one hand, the SOC decline visible on the time series might repudiate the 4 per mille concept, on the other, the lower decrease observed under RI+PM might suggest that the practice might increase, or better, decrease less compared to the standard practice. To the best of our knowledge, the latter approach, namely using the 4 per mille for comparing treatments time by time, is the approach commonly used in literature. We wish to include both discussion points in our revised paper by including also a paragraph repudiating the 4 per mille at our site.

- Thoroughness and logic of various data interpretations. The authors find no effect of adding poultry manure on SOC, which is surprising given that exogenous C sources have previously been found to be highly effective in increasing local SOC levels. However, the rate of poultry manure dry matter addition is much lower than normally studied (1 Mg / ha annually; in meta-analysis of Kallenbach et al 2011 doi:10.1016/j.agee.2011.08.020, 5 Mg / ha is lowest dry matter addition category), which is not currently contextualized. The quantity of poultry manure was originally set to balance the C/N ratio for improving crop residue humification. For more clarity, we will include a sentence in the text. Near L200, the authors claim that the experimental site was not a SOC equilibrium, "confirmed by the low C/N ratios"; how can C/N ratios be used to infer a non-equilibrium state? The C/N ratio is used as a proxy to infer the soil C dynamics and very low values might indicate high mineralization activity with pauperisation of C stock. However, we can remove this sentence to improve the text's readability. Near L250, the authors claim how, in general, calcaric, silty soils are inert to management practices because they are compacted and sometimes anoxic, but how are any of these characteristics grounds for a soil being unresponsive to management practices? Those characteristics are not directly responsible for soil inertia, we will clarify this sentence.

- Near L255, the authors claim that SOC levels declined in the study as a result of 'agricultural intensification', but different levels of agricultural intensification were not clearly compared in the study, so how could this statement factor so prominently in their conclusions? This statement results from what is speculated in the discussion (LL196-201) "The experimental farm where the LTE is located was owned by the University by 1962. Before the start of the experiment, the field was conducted following the intensification of agricultural practices, typical of the first part of the 60s" [...] "Therefore, the general SOC decline might not represent the effect of a land-use change but rather a movement from a threshold level to another inside a cultivated agroecosystem, in particular a shift from a low (e.g., shallow non-inversion tillage) to more intensive agriculture (e.g., moldboard ploughing)."

  Writing style. The manuscript writing, in terms of structure and style does not yet meet a high standard of quality. Paragraph structure is not consistently used, as some sentences are presented outside of paragraphs (last sentence of introduction; second sentence of section 2.2; last sentence of section 3.1). The meaning of 'sensibly' as a modifier, repeatedly used, is unclear. Parts of the manuscript are written in an informal or casual style that should be revised in order to be suitable for publication in a scientific journal ("Anyway", "Nowadays").

  We thank the reviewer for pointing out this issue, we will use an international language service before the revised paper re-submission.

---

## Author Comment (AC2)

General Comments –

The manuscript submitted by Piccoli et al. presents the results of a 54 year long field experiment, with a focus on soil carbon accumulation in response to residue management treatments. Using mixed models, the authors attempt to understand how residue incorporation drives soil C and N stocks and concentrations, and then apply a series of empirical models for determining soil C saturation to explain their findings. Additionally, they examine the time-series of data stretching back to 1966 to understand the long-term impacts of the residue management treatments. The authors find that residue removal did significantly decrease SOC stocks relative to residue incorporation treatments, though the general trend of the SOC stocks since the experiments inception has been a loss. They find little evidence of saturation across the models they employ. Overall, the authors present an interesting and important dataset that should continue to be investigated, as long-term SOC datasets continue to be sparse. There are several issues that bar the manuscript from publication in its current form, however. The writing style is often overly casual, and does not adhere to typical paragraph form, making the reading of the MS somewhat confusing and choppy. In addition, the methods are not plainly described, leading to a difficulty on the part of the reader to evaluate the authors results. This is most relevant in the discussion of sampling, the calculations of the saturation values, and the rationale for the statistical tests that were employed. The results section is relatively brief, while some data, such as the time-series results, are not introduced at all until the discussion. Other data that are relevant the the conclusions drawn, such as the impact of N rate or the interaction effect between N rate and residue treatment, are not presented at all. The concluding statements of the manuscript introduce new concepts as well, not previously discussed throughout the paper. Finally, the MS is either incomplete or contains errors, as Table 1 is not included in the text, despite in-text references. I've detailed my specific comments below.

Specific Line Edits-

Lines 28 – 30: Do you think this assertion holds up across all ecosystems? I agree that the mineral-associated fraction has a finite upper limit on its accumulation, but see Cotrufo et al. 2019 for evidence of particulate organic matter accumulation that lacks a saturation limit. We thank you for suggesting such an interesting paper that will be promptly included in the revised MS version.

Lines 39-43: There's been a lot of recent work on saturation behavior in soil carbon (e.g., Georgiou et al., 2022; Heckman et al., 2023) that is relevant to the present study. I recommend the authors broaden the scope of their literature review in this section to include both the foundational and contemporary literature. We thank you for suggesting such interesting papers that will be promptly included in the revised MS version.

Line 53: There was a rapid shift in subject here that was hard to follow as a reader. I recommend the authors consider restructuring this paragraph to better communicate the ideas presented. We will revise the text, as suggested.

Lines 89-91: Please describe the overall tillage management for each plot here. It is currently unclear how tillage is handled between treatments – does the residue removal treatment receive the same level of cultivation as the incorporation and manure treatments? If not, please describe how this has been accounted for in the resulting analysis. Yes, the reviewer understood correctly as all plots are moldboard ploughed at the same time. We will specify better that all plots receive the same cultivation intensity.

Lines 99 – 103: I found the description of sampling here to be confusing. Two things in particular stand out: First, did the authors collect one sample per plot? Right now, that is not clear from the text, which implies that 60 samples were taken per plot. Second, were two sets of samples taken, one for bulk density and one for elemental analysis? Or was the original sample used for both analyses? We thank the reviewer for pointing out this issue. We will modify the entire sampling section clarifying the sampling procedure. In brief, the soil sampling involved two different samplings. 1) Undisturbed soil cores (7 cm diameter, 60 cm height) were collected from the middle of each plot using and hydraulic sampler, cut into distinct layers (0-30 and 30-60 cm) and oven-dried for bulk density determination. 2) A series (4/5) of remoulded soil sampling were collected from different parts of the plot (avoiding borders) to form a single bulk soil sample for each plot and sampling depth (0-30 cm and 30-60 cm) using a hand-push auger. Afterwards, the 120 (60 plots x 2 depths) soil samples were air-dried and subjected to chemical analysis.

Line 111: I believe that the experimental design the authors have described is a randomized complete block, split-plot design, which residue management the main plot effect, and N rate treatment as the split-plot effect. Did the authors consider this when designing the error structure of their mixed model? More directly, shouldn't there be an additional random effect that specifies the nested structure of the design and specifies random intercepts for the block/residue/N rate combination? We thank you for the comment, we performed an ANOVA for the split-plot design with the residue management as the main plot, N rate as the sub-plot and block effect as random. We will clarify better the text and check through the entire manuscript for consistency.

Line 120: As mentioned above, several recent papers have provided updated means for understanding soil C saturation and saturation deficit (Georgie et al., 2022 in particular). Please provide some rationale for the use of the Hassink and Dexter models in lieu of the more contemporaneous approaches, either in response or in the text. We arbitrarily decide to use the most used (cited) models. The number of citations does of course not necessarily means the paper is better than another one but this is a standard practice literature review. We thank you for the paper suggestion, unfortunately with only "Georgie et al., 2022" and any other identifier (e.g., doi) we cannot find any relevant paper with that name. If the reviewer refers to Georgiou et al., 2022 10.1038/s41467-022-31540-9 please see the reply to Reviewer #1 comments.

Lines 123 – 125: Please show the equations for the saturation models that you employ here such that they can be referenced during the reading of the results.

Ok, we can add the equations during text revision.

Line 136: Is there an established agronomic optimum N rate for this site? If so, does it inform at all the relationship between SOC and N rate? The studied N range (0-240 kg N/ha/y) covers the commonly used N fertilization level in the area which depends on crop type.

Lines 170 – 175: Is there a description of how the authors either isolated or estimated the different soil particle size class distributions used in these models? Please describe this process and include in the methods section. The soil particle class were analysed with laser diffraction and converted into pipette using the algorithm reported in Bittelli et al., 2022 https://doi.org/10.1016/j.geoderma.2021.115627. We will add this information to the text.

Line 184: None of the time-series data is presented in the Results section, which makes its introduction here somewhat surprising and confusing. I encourage the authors to add these data and the results that they glean from it to the results section prior to discussing it here. We understood the Reviewer's point of view and we will add those data descriptions in the results section.

Line 195: I'm confused as to why the x-axis on this plot extends to 2030. If the authors aren't projecting out to that date, I suggest that reformat the axis to represent the data that they have available and are presenting.

Ok, we can modify the x-axis.

Lines 198 – 199: Please provide further explanation as to your assertion here, that a lower CN ratio in 1966 is evidence that the system was out of equilibrium. Generally, very low C/N value refers to high mineralization activity, however, we can remove this half sentence for more clarity.

Lines 209 – 211: I'm a little confused here – are the authors suggesting there is a biophysical limitation on the ability to maintain carbon stores over time? Unless I'm misinterpreting, does this imply that agricultural systems are restricted from reaching a steady state, even over the course of decades? Agricultural systems usually reach a steady state only after decades (e.g., more than 30 yr). What we would stress here is that, as reported by Berthelin et al., 2022, the practice of adding C input with agricultural biomasses is not so efficient because requiring about 10 times the weight of what is expected to be sequestered as SOC. We will re-phrase the sentence for more clarity.

Lines 224 – 228: This is an interesting point, but I'm not sure how it's relevant to the question at hand regarding the potential of reaching the 4 per mille goal via residue incorporation. I recommend the authors make the linkage more explicit here. Further, I'm

not sure this accounting is fully inclusive. Given the results from above, that over the course of the experiment residue removal has lost significantly more carbon that residue incorporated treatments, would you still arrive at this results if you factored in the lost soil C in addition? I could be misunderstanding the math the authors use here, but I would appreciate their clarification. We got the point of the reviewer. The use of crop residue for bioenergy production should have saved 0.74 t C ha/ y x 50 years = 37 t C/ha while their incorporation led to a delta of ca. 4 t SOC /ha, 10-fold more CO2-equivalents than SOC accumulation by incorporation. A correct comparison between different crop residue usage would be necessary, anyway, we wish to stress that different types of use of residue can be conceivable.

Lines 232-233: Please clarify this statement, as in lines 135 and 154 the authors state that N rate does affect both C concentrations and stocks. We will clarify better this paragraph.

Line 244: Where is Table 1? The authors have not included it in the present manuscript, limiting the interpretation of the results they discuss here. The reviewer is right, we will add the table in the revised version.

Line 264 – 267: The authors here are presenting the discussion of carbonates as though this is a central finding from the manuscript, but this is the first time that it is discussed. Additional text is needed in the discussion section for the reader to have the context necessary to interpret this. We will add additional text, as suggested.